# Internal and External Validity of Social Media and Mobile Technology-Driven HPV Vaccination Interventions: Systematic Review Using the Reach, Effectiveness, Adoption, Implementation, Maintenance (RE-AIM) Framework

**DOI:** 10.3390/vaccines9030197

**Published:** 2021-02-26

**Authors:** Matthew Asare, Braden Popelsky, Emmanuel Akowuah, Beth A. Lanning, Jane R. Montealegre

**Affiliations:** 1Department of Public Health, Baylor University, Waco, TX 76708, USA; braden_popelsky@baylor.edu (B.P.); emmanuel_akowuah@baylor.edu (E.A.); Beth_Lanning@baylor.edu (B.A.L.); 2Dan L Duncan Comprehensive Cancer Center, Baylor College of Medicine, Houston, TX 77030, USA; jrmontea@bcm.edu

**Keywords:** HPV, HPV vaccine, social media, mobile phone, HPV vaccine intervention, RE-AIM Framework

## Abstract

Social media human papillomavirus (HPV) vaccination interventions show promise for increasing HPV vaccination rates. An important consideration for the implementation of effective interventions into real-world practice is the translation potential, or external validity, of the intervention. To this end, we conducted a systematic literature review to describe the current body of evidence regarding the external validity of social media HPV vaccination-related interventions. Constructs related to external validity were based on the reach, effectiveness, adoption, implementation, maintenance (RE-AIM) framework. Seventeen articles published between 2006 and 2020 met the inclusion criteria. Three researchers independently coded each article using a validated RE-AIM framework. Discrepant codes were discussed with a fourth reviewer to gain consensus. Of these 17 studies, 3 were pilot efficacy studies, 10 were randomized controlled trials (RCTs) to evaluate effectiveness, 1 was a population-based study, and 3 did not explicitly state which type of study was conducted. Reflecting this distribution of study types, across all studies the mean level of reporting RE-AIM dimensions varied with reach recording 90.8%, effectiveness (72.1%), adoption (40.3%), implementation (45.6%), and maintenance (26.5%). This review suggests that while the current HPV vaccination social media-driven interventions provide sufficient information on internal validity (reach and effectiveness), few have aimed to gather data on external validity needed to translate the interventions into real world implementation. Our data suggest that implementation research is needed to move HPV vaccination-related interventions into practice. Included in this review are recommendations for enhancing the design and reporting of these HPV vaccination social media-related interventions.

## 1. Introduction

The human papillomavirus (HPV) vaccine protects against HPV-associated cancers, including most cervical cancer, as well as vulvar, vaginal, anal, penile, and oropharyngeal cancer. Cervical cancer is the fourth most common cancer among women worldwide, with approximately 570,000 new cervical cancer cases reported in 2018, representing 6.6% of female cancers [1]. The incidence of oral and anal cancers is increasing [2,3]. The HPV vaccine is recommended for adolescents aged 11–12 years, with catch-up vaccination through age 26 and FDA approval for adults up to age 45 years [4]. As part of the Global Strategy for the Elimination of Cervical Cancer as a Public Health Problem, the World Health Organization’s (WHO) goal is for 90% vaccination of girls age 15 by 2030. The Healthy People 2030 goal is to increase the proportion of adolescents who receive recommended doses of the HPV vaccine with a target goal of 80% [5]. Although integrated programs and efforts to increase the HPV vaccination have occurred in many countries over the last 14 years, HPV vaccine rates remain low [6]. For instance, in 2019, only 54.2% of adolescents in the US [7] and 15% of adolescents globally [8] were current on HPV vaccinations.

Multiple factors including lack of opportunity for vaccination, parental attitudes or perceptions towards vaccination, lack of recommendations from healthcare providers, concerns about the vaccine’s effect on sexual behavior, religious objection, low perceived risk of HPV infection, social influences, irregular preventive care, and vaccine cost have contributed to the low vaccination rates [9,10,11]. In efforts to address these barriers, researchers have included mobile technology-related media including text-message, e-mail, phone calls, and private Facebook messages, in their interventions to increase vaccination awareness, uptakes, and dose completion [12,13]. These interventions have been efficacious in increasing HPV vaccination uptake and completion [12,13,14,15]. However, the prospect of translating these efficacious interventions into regular clinical practice is unknown due in part to the lack of reported external validity [16]. External validity is the ability to generalize an evidence-based study to different measures, persons, settings, and times [17]. Several translational researchers have argued that reporting detailed components and processes of evidence-based studies would increase studies’ generalizability (external validity) and the ability to translate those interventions into practice [18,19,20].

To address the research-practice issue, Glasgow and colleagues developed the Reach, Effectiveness, Adoption, Implementation, and Maintenance (RE-AIM) framework [21] with a set of metrics critical for evaluating the generalizability of an evidence-based intervention into routine practice. They proposed that the translatability of an intervention is best evaluated through the five dimensions (RE-AIM). The reach dimension is designed to assess the proportion of potentially eligible individuals who participate in the intervention study. Efficacy/effectiveness is the function of the intended positive impact of the intervention. Adoption reflects the potential settings and intervention agents that participate in a study. Implementation refers to the quantity and quality of delivery of the intervention’s various components. Finally, the maintenance dimension is the longer-term efficacy/effectiveness of an intervention on an individual (see Table 1 for the definitions and indicators for each RE-AIM dimension). This framework can be used to organize and evaluate threats to the transferability of research to practice. Further, by evaluating a study through the five-dimension lens of the framework, researchers are able to assess internal and external validity equally [16,21].

The RE-AIM framework has been used recently in several systematic reviews to evaluate the internal and external validities of health intervention studies such as weight management intervention [18], physical activity intervention [22,23], worksite health behavior interventions [24], community settings [20], school-based health promotion [25,26], childhood obesity prevention [27,28,29], children dietary interventions with parents [30], injury prevention strategies [31], faith-based intervention [32], mobile phone-based intervention for diabetes self-management [33], and HIV prevention intervention [34]. While encouraging, there is little reporting on its potential use for translating HPV vaccination social media-driven intervention methods into regular practice settings, specifically at the population level. The two most recent comprehensive literature reviews on HPV vaccination mobile technology-related interventions evaluated the effectiveness of interventions in increasing the HPV vaccine [35], and the effectiveness of communication technology interventions on HPV vaccine [36]. However, the scope of these two previous literature reviews was narrow, focusing primarily on internal validity with limited information on the external validity of the HPV vaccination intervention studies [18,19].

Our study aimed to use the RE-AIM framework to evaluate HPV vaccination intervention studies that included mobile technology to increase HPV vaccination completion (i.e., receiving all the recommended doses: two doses for 9–14 years old and three doses for those between 15 and 45 years old) and/or vaccination uptake (defined as receiving at least one dose of vaccine). To our knowledge, this is the first review using the RE-AIM framework to evaluate the HPV vaccination mobile technology-related (Facebook, text messaging, and mobile health (miHealth)) interventions. Unlike the two previous reviews on HPV vaccine interventions [35,36], our current systematic review was structured to determine the translation potential or external validity of published HPV vaccination intervention studies by determining the extent to which those studies reported information across all five of the RE-AIM framework dimensions. We further provide recommendations for future research based on these findings.

## 2. Materials and Methods

### 2.1. Search Strategy and Selection Criteria

We conducted an extensive literature search to identify research articles related to HPV vaccination technology-based interventions. We searched nine databases (PubMed, EMBASE, Medline, ERIC, CINAHL, Academic Search Complete, Web of Science, PsycINFO, Cochrane Library) using the following terms: human papillomavirus OR human papillomavirus* OR HPV and social media OR social medium OR Web 2.0 OR twitter messaging OR Instagram OR Facebook OR WhatsApp OR Tito OR text message OR mobile technology AND intervention OR RCT. An article was included in the review if it met the following inclusion criteria: published in English, between 2006 and 2020, and in peer-reviewed journals; outcome variables included HPV vaccination completion and/or vaccination uptake; intervention study; the intervention’s mode of delivery included social media (WhatsApp, Facebook) and text messages. Articles were excluded if they were cross-sectional studies and included assessment of only participants’ knowledge, attitude, and intention (see Figure 1).

### 2.2. RE-AIM Criteria

A modified version of the RE-AIM 30-item data extraction tool (https://www.re-aim.org, accessed on 10 February 2021) was used to code eligible articles on the degree to which internal and external validity indicators were reported. The RE-AIM dimensions and corresponding indicators are listed in Table 1.

#### 2.2.1. Reach Dimension

Seven indicators were used to evaluate the reach dimension of the study. They included the description of the target population, description of the participants’ HPV vaccination behavior, recruitment strategies, inclusion and exclusion criteria, description of the sample size determination, and participation rate.

#### 2.2.2. Effectiveness Dimension

Eight metrics for effectiveness included the efficacy of the intervention in changing vaccination behavior, measurement of primary and/or secondary outcome (i.e., vaccination completion, or uptake), a short-term assessment, intent-to-treat assessment, description of imputation procedure, the measure of robustness across subgroups, and short-term attrition assessment.

#### 2.2.3. Adoption Dimension

The seven indicators used for adoption were a description of intervention location, intervention delivery staff, the method used to identify delivery staff, inclusion and exclusion criteria for the staff, and rate of staff participation.

#### 2.2.4. Implementation Dimension

The four metrics used for the implementation dimension were intention frequency, duration, the extent to which the intervention was implemented as planned, completion rates, and measurement of cost of implementing the intervention.

#### 2.2.5. Maintenance Dimension

The four metrics for maintenance included follow-up (3 and 6 months) assessments, attrition rates, continuation, and institutionalization of the program.

### 2.3. Coding and Analysis

Articles that met the inclusion criteria for this review were independently coded by three graduate research assistants (B.P., N.T., and C.M.) and supervised by the principal investigator (PI) of the research team (M.A.). Each reviewer coded a “yes” indicating the presence or “no,” indicating the absence of the RE-AIM indicators outlined above. Following the individual coding, the PI and the three research assistants met to discuss articles and coding results, resolve uncertainty, and gain consensus in coding. The articles that had “yes” for any indicator were scored as 1 and “no” was scored as 0. Each of the 17 articles was tabulated and scored with a column totaling individual dimension scores for each article. Analyses included providing count and percentage data across RE-AIM indicators. Row percentages were calculated to display the proportion of articles addressing each of the dimension indicators. Finally, column totals, averages, and average percentages were computed to summarize the number of articles reporting each of the five dimensions. To determine the overall quality of RE-AIM reporting, we also examined the number of articles that included the 33 indicators from the data extraction tool (see Table 1). Based on the 33-item RE-AIM indicators, articles that scored between 0 and 11 indicated fewer reporting of RE-AIM indicators, between 12 and 22 indicated moderate reporting, and between 23 and 33 indicated high reporting of RE-AIM indicators. 

## 3. Results

The literature search yielded 414 total articles. After removing duplicate articles and conducting a preliminary screening process based on title and abstract, a total of 53 eligible articles remained. The secondary screenings restricted articles to HPV vaccine interventions and social media-driven interventions. The complete article identification strategy produced 17 articles that met the inclusion criteria and were analyzed in this review [15,37,38,39,40,41,42,43,44,45,46,47,48,49,50,51,52]. Of these 17 studies, 3 were pilot efficacy studies [39,44,51], 10 were randomized controlled trials (RCTs) to evaluate effectiveness [15,38,43,45,46,47,48,49,50,52], 1 was a population-based study [41], and 3 did not explicitly state which type of study was conducted [37,40,42]. The target population for the reviewed articles includes parents with adolescents (boys and girls) [15,45,46,50], Young Sexual Minority Men [51,52], college students [37,38,44,47,48,49], adolescents (boy and girls) [40,41,42], young women (19–26 years) [39,43]. The vaccination uptake rates in the review ranged from 6.6% to 89% and the completion rates range from 17% to 88%, indicating successful implementation of many of the interventions. A total number of 189,877 participants were reached in the reviewed HPV vaccine interventions. Out of 189,769, (we excluded 108 participants in Ortiz et al.’s study because they did not provide absolute numbers for vaccination uptake) participants enrolled in the re-viewed articles, pooled estimate of 19,294 participants in studies received at least one dose of HPV vaccine representing a 10.2% vaccination rate among the participants in those reviewed articles. Mohanty et al. [41] and Chodick [50] used Facebook to deliver the intervention and they reached the largest target population or had highest penetrations (155,110 and 21,592 participants, respectively) and text messaging interventions. The most common social media used were mobile phone text messaging [39,46,49,52], combination of text messaging and email systems [15,43,47,48], Facebook [41,42,50], and mobile web technology [38,40]. Other studies mentioned that they used social media but did not mention specific social media [37,44,51]. Appendix A shows the details on study design, outcome, demographic characteristics, social media used, and RE-AIM indicators used. 

### 3.1. RE-AIM Reporting Scores

Using RE-AIM rating procedures, we found that 14 (82.35%) articles moderately reported RE-AIM indicators [15,37,39,40,41,42,43,44,45,46,48,50,51,52], and three articles (17.65%) had high reporting of RE-AIM indicators [38,45,47,49]. The three studies that were rated as high quality addressed between 69.70% (23 out of 33) and 72.73% (24 out of 33) of the indicators. The 14 medium reporting quality articles addressed between 42.42% (14 out of 33) and 66.67% (22 out of 33) of the indicators. The reach and effectiveness dimensions were the most addressed domains with an average score of 90.8% and 85.8%, respectively. The adoption, implementation, and maintenance domains were the least addressed domains with average scores of 38.2%, 45.6%, and 26.48%, respectively (see Table 1).

#### 3.1.1. Reach Dimension

The indicators with the greatest reporting scores under the reach dimension were description of the target population, including race/ethnicity and other demographic information (100%), behavioral information (100%), inclusion/exclusion criteria (94.1%), and recruitment strategies (94.1%). The least reported reach indicator was the method to identify the target population with an average score of 76.5% (see Table 1). Overall, ten articles [15,38,41,42,46,47,48,49,51,52] reported all seven indicators under the reach dimension, four studies [37,40,44,45] reported the least (5 out of 7) of the indicators in the reach dimension (see Figure 2).

#### 3.1.2. Efficacy/Effectiveness Dimension

The most common efficacy/effectiveness dimension indicators reported by the reviewed articles include intervention effectiveness in changing behavior (100%), measurement of the primary outcome (100%), short-term assessment (94.1%), and description of intervention design/conditions (100%). Indicators such as intent-to-treat, imputation procedure, quality of life, unintended consequence measurement, a measure of robustness across subgroups, and measures of short-term attrition were scarcely reported in the articles (see Table 1). Three articles reported eight out of 10 effectiveness dimension indicators [38,43,47]. Fontenot et al. [51] and Mohanty et al. [41] reported the least amount, 4 out of 10 indicators (see Figure 2).

#### 3.1.3. Adoption Dimension 

For the adoption dimension, the most reported indicators were the description of intervention location (88.2%) and the staff who delivered intervention (64.7%). Organizational spread and measures of the cost of adoption were two adoption indicators that received the lowest reporting score of 5.9% (see Table 1). Only Gerend et al.’s [37] article reported six out of the eight adoption dimension indicators. The remaining articles reported a few of the adoption dimension indicators with scores ranging from 1 to 5 (out of 8 indicators) (see Figure 2).

#### 3.1.4. Implementation

Per our inclusion criteria for this review, all 17 articles included in the review used some form of technology including social media (Facebook), mobile phone (text messages), and emails to deliver the intervention. The most addressed implementation dimension indicators were intervention frequency and duration (88.2%) and participation/completion rates (82.4%) (see Table 1). The extent to which the protocol was delivered as intended was the least reported indicator (11.8%) and none of the studies reported the total cost of implementing the intervention. However, eight reported incentives given to each participant, and one reported that vaccines were given to participants at no cost (see Figure 2).

#### 3.1.5. Maintenance

The most common maintenance dimension indicator addressed in the reviewed articles was the 3-month and 6-month follow-up assessment (70.6%). The long-term attrition rate was addressed in 35.3% of articles. None of the articles addressed whether the programs were institutionalized or were still in place (see Table 1).

## 4. Discussion

### 4.1. Summary

This systematic review utilized the RE-AIM framework to evaluate the impact of HPV vaccine intervention studies that incorporated mobile technology to increase vaccine uptake and recommended dose completion. Eleven articles’ outcome of measure included vaccination uptake [37,38,39,41,44,46,47,49,50,51,52], and six articles’ outcome measure was vaccination completion [15,40,42,43,45,48]. The vaccination uptake rates in the review ranged from 6.6% to 89% and the completion rates range from 17% to 88%, indicating successful implementation of many of the interventions.

Our review of 17 articles showed the emphasis on reporting internal validity (i.e., reach and efficacy), and the collective absence of reporting external validity per Glasgow et al.’s (1999) reporting criteria. This finding is consistent with other RE-AIM literature reviews that found that the most common indicators reported in studies are reach and efficacy/effectiveness [23,25,26,27,28,32,53]. On the other hand, the external validity dimensions which include adoption, implementation, and maintenance were underreported. Therefore, making the translation of those intervention studies into the practice setting difficult.

#### 4.1.1. Reach Dimension

A total number of 189,877 participants were reached in the reviewed HPV vaccine interventions. Lee et al.’s [39] article reached the lowest number of participants (*n* = 30) while Mahanty et al.’s [41] Facebook intervention reached the highest number of participants (*n* = 155,110). The reporting for the reach dimension indicators ranges from 77% to 100%. Many of the articles included in this review provided detailed descriptions of the target population which is consistent with the literature [25,27,28,29,30,34,54,55]. Several articles targeted college students [37,38,43,44,48] which is not surprising given that the majority of those within the college-age groups are social media consumers. A few of the articles reviewed reached or targeted parents with adolescents (boys and girls) [15,45,46,50]. Parents are critical target audience for any HPV vaccination interventions because parental knowledge, positive attitude, affordability, and willingness are precursor to successful HPV vaccination programs [56,57,58,59]. The reach dimension indicator rarely discussed is the target population denominator. While reporting of this indicator tends to be challenging [55], by not reporting the target population denominator, there is no context given to help determine the sample sizes. In the efficacy studies the concern is recruiting enough participants to provide the necessary power to detect effect size; therefore, understanding the target population that was exposed to recruitment materials can provide an estimate of the likely reach the program will achieve [18].

#### 4.1.2. Efficacy/Effectiveness

Out of 189,769 participants enrolled in the reviewed articles, a pooled estimate of 19,294 received at least one dose of HPV vaccine, representing a 10.2% vaccination rate among the participants in those reviewed articles. Tull et al.’s [49] study reported the highest vaccination rates (motivational arm 88%, self-regulatory arm 89%, and control arm 86%). Our systematic review showed that four indicators in the Efficacy/Effectiveness dimension including design/conditions, efficacy or effectiveness, the measure of the outcome with or without comparison to the vaccination goal, and short-term assessments, were regularly reported which is consistent with the literature [25,27,28,29,30,34,54,55]. Across all 17 articles, less than half the authors described their chosen method of analysis for missing variables and/or attrition whether to use intent-to-treat or per-protocol analysis (analysis by treatment administered) approach. The intention-to-treat principle states that all randomized participants are included in the statistical analysis and analyzed according to the group they were originally assigned, regardless of what treatment (if any) they received [60]. Whenever treatment groups are not analyzed according to the group to which they were originally assigned, the risk of bias increases [61]. Similar to the conclusion drawn by Hollis and Campbell [58] the intention to treat approach is often insufficiently described and inadequately applied. This lack of reporting can diminish the accurate (unbiased) conclusions regarding the effectiveness of an intervention [61]. Authors should explicitly describe the handling of any deviations from randomized allocation and discuss missing responses and their potential effect on the studies’ outcomes [62]. However, the use of randomization in some of the studies [15,38,40,42,43,45,52] may attenuate the possible confounding bias. Although most of the articles reported short-term pre-post assessments, only four articles [38,43,45,53] calculated short-term attrition rates. Attrition prevents a full intention to treat analysis and it can occur when participants have missing data and/or loss to follow-up. We argue that researchers need to be more explicit about the loss to follow-up, especially if rates are high [63].

#### 4.1.3. Adoption Dimension 

The average score of the adoption dimension was 38% with observed scores for the adoption indicators ranging from 6% to 88%. The description of the location of the intervention score (88%) for our reviewed articles is higher than the previous reviews scores of 48% to 60% [18,54], but lower when compared with the scores in Allen et al.’s [16] review findings. The description of the delivery staff received the second-highest in reporting (65%). The skill sets of the intervention staff, the description of the setting of the intervention staff, and information about the staff can help determine the translating potentials of the study in another setting. Less than half of the articles discussed the methods used to identify target delivery agents and the level of expertise. Crucial to the implementation of an intervention is the selection of the target delivery agents but indicators such as inclusion and exclusion criteria for selecting delivery staff, delivery staff participation rate, organizational spread, and measure of the cost of adoption were underreported. The underreporting of selection criteria of delivery agents has been debated by other reviews [16,27]. By not reporting a level of expertise or a method to select the interventionists, it becomes difficult not only to measure the intervention effectiveness but also to translate those studies to other settings to achieve the same level of success. Rarely reported is the organizational spread dimension indicator, which measures if the intervention used a multi-level approach. This is an important indicator because of the focus on collaboration and communication between multiple levels of an organization which can increase the overall impact of the intervention and the probability of the behavior being adopted and/or maintained. The full support of adopting agents directly influences implementation fidelity and program sustainability [64].

#### 4.1.4. Implementation

In the implementation dimension, we were interested in intervention studies that incorporated social media technology to deliver the intervention. Although the authors did not evaluate whether using a specific social media type was effective in delivering the intervention, based on our review we found that interventions that utilized Facebook [41,50] and text messages [15,45,49] reported significant improvements in HPV vaccination. The most frequent technology used was text messaging [15,37,39,43,45,46,47,48,49]. Additionally, only eight articles reported the underlying theoretical framework used in intervention development [38,39,41,42,49,50,51,52]. Theories provide a systematic view of phenomena by specifying the relationship between program inputs (resources), program activities (how the program is implemented), and their outputs or outcomes [65,66]. Reporting the theoretical framework used in intervention could facilitate replication and implementation of HPV intervention studies in several different settings.

The HPV vaccine intervention studies included in this review required multiple visits and multiple contacts (reminders) with the study population. Omitting information about the number of contacts or the exact reminder messages introduces a bias. The types of reminders and the difficulty in reaching vaccination sites can greatly influence vaccination uptake. Additionally, the impact of the intervention is weakened by the lack of reporting on the extent to which the protocol was delivered as intended, including sending reminders to participants. Furthermore, none of the articles included the costs incurred during the implementation phase of the interventions, which once again eliminates a monetary reference point to consider when designing future interventions. While many researchers incorporated incentives for participation and offered vaccines at reduced or no costs, there is minimal discussion of the cost incurred in implementing the interventions. Reporting cost is essential in understanding how resources are utilized in both effective and non-effective interventions. Cost is a critical aspect of interventions designed for low resource areas, as the success of those interventions is partly dependent on study participants’ ability to pay for the cost of the vaccination.

#### 4.1.5. Maintenance

Our review showed the maintenance dimension was by far the most underreported indicator, with only 68.75% of articles reported conducting follow-up assessments. The lack of reporting could be due to researchers’ financial and time limitations [20]. For HPV vaccines, it requires 12 months or more to complete the recommended dosing. HPV intervention studies should include a measurement of long-term effects and address the issue of maintenance. Without such measures, it is difficult to determine if the intervention strategies affect vaccination completion rates.

### 4.2. Limitations

There were several limitations to our study. First, this review focused on increasing the uptake of the HPV vaccine series. Because the vaccine comprises two to three doses administered at varying intervals in large catchment periods and populations, follow-up and series completion can be inherently difficult to ensure. Second, the RE-AIM framework served as the guide for evaluating the effectiveness of public health interventions. However, we utilized a modified or shortened version of the RE-AIM criteria in each dimension. Third, it is possible that our scope for article selection was significantly narrowed because we concentrated on reviewing recent articles involving a mobile technology and social media component through electronic messages/reminders, and/or social media campaigns. Fourth, while the interventions targeted several different populations, (e.g., college students, Appalachian women, adolescents, urban, rural, etc.) three of the articles in the review took place outside the United States with different contextual factors. Public health attitudes, perceptions, resources, and procedures may differ across cultures. Finally, researchers of the reviewed articles may argue that their studies were intended to demonstrate the efficacy of the intervention and that the scope of reporting might preclude effectiveness or generalizability information [20,67]. Further, researchers may collect data on external validity, but due to article length restriction may not report the date. However, even in efficacy trials, it would be beneficial to document and report adoption and implementation dimensions so that future researchers can replicate the study [20,67].

### 4.3. Strengths

Despite the above limitations, this review has several strengths. First, the use of the RE-AIM framework as a guiding metric for evaluating HPV vaccination interventions. The RE-AIM framework allows for the inclusion of the internal and external validity criteria that are important for evaluating the possibilities of translating interventions to other settings instead of evaluating just the efficacy of the interventions. Second, RE-AIM offers a systematic and structured examination of system-level considerations to the adoption of efficacious interventions. The RE-AIM framework has shown utility in assessing multiple criteria related to prevention research, namely, elements of efficacy, effectiveness, efficiency, and other implications for public health decision making such as quality of life and safety [64].

### 4.4. Implications and Recommendations

#### 4.4.1. Implication for Future Publication

The application of the RE-AIM framework to evaluate the effectiveness of HPV vaccine social mobile interventions is limited, yet our review demonstrates the utility of RE-AIM to evaluate HPV interventions and highlights the potential transferability of selected HPV intervention programs to a broader audience. To increase the potential to translate social media related HPV vaccine research findings to practice, researchers should place a greater emphasis on obtaining and reporting external validity information, such as adoption, implementation, and maintenance dimensions. Providing external validity information enhances other researchers’ and practitioners’ ability to judge the generalizability of effects and the comparative utility of interventions [20,67]. All stakeholders, including researchers, reviewers, editorial board, and funders, should emphasize the need for external validity information [20].

#### 4.4.2. Implications for Future HPV Vaccine Intervention

While a few reviewed studies included parents of adolescents [15,45,46,50], there is a need to consider social media strategies as a potential method to reach parents. Parents either make decisions to vaccinate their teenagers or influence their children’s decisions so not including them in the target population is a missed opportunity to influence behavior [68].The overall penetration or reach of the studies was high, especially in studies that used Facebook to reach a large population [41,50]. However, the impact of social media on the vaccine uptake was rarely measured in the reviewed studies. Future studies should compare the effectiveness of different social media platforms (e.g., Facebook vs. text messaging) on HPV vaccine uptake.

## 5. Conclusions

In conclusion, our findings show that social media-related HPV vaccination intervention studies demonstrated some effect on vaccination uptake (at least one dose of vaccination rate of 10.2% of the study population), reached larger study participants, and demonstrated that college students and college-aged groups are the targets of most social media intervention studies. While most articles in our review met the Consolidated Standards of Reporting Trials (CONSORT) metrics for reporting, specifically for internal validity reporting, the adoption, implementation, and maintenance dimensions for the RE-AIM framework were underreported. To ensure that these successful community-based interventions can be translated into practices, stakeholders should not only embrace the reporting of all the RE-AIM dimensions but should encourage researchers to adhere to external validity reporting standards similar to CONSORT internal validity reporting.

## Figures and Tables

**Figure 1 vaccines-09-00197-f001:**
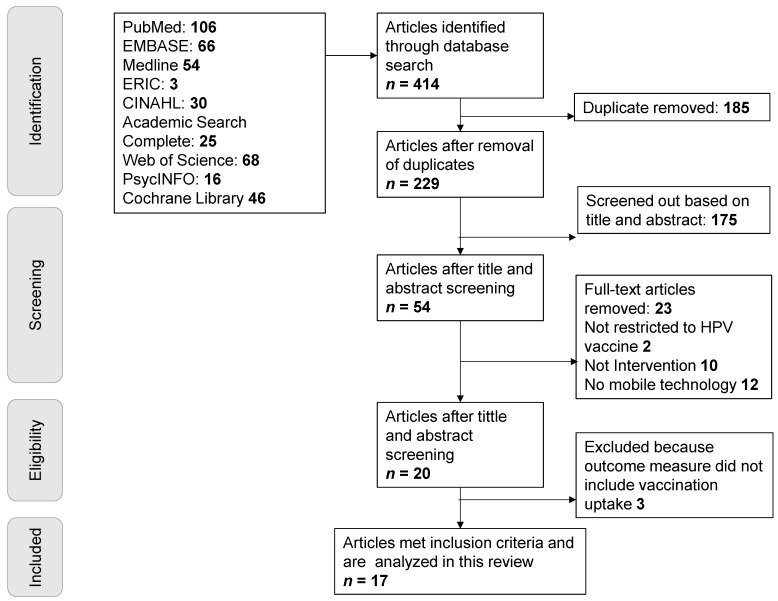
Flowchart of the search strategy.

**Figure 2 vaccines-09-00197-f002:**
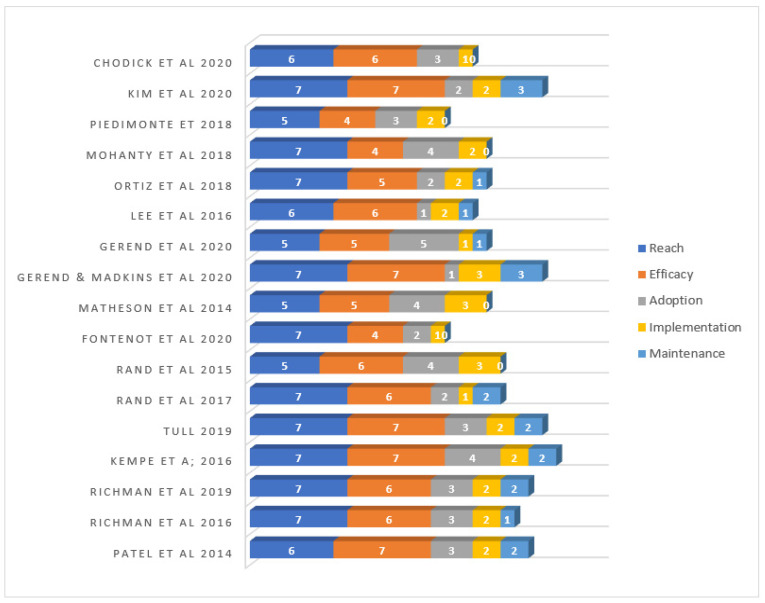
The scores for the reviewed articles on the RE-AIM Dimensions (*n* = 17). Note: Reach scale 0–7; Efficacy scale 0–10; Adoption scale 0–8; Implementation and Maintenance scales 0–4.

**Table 1 vaccines-09-00197-t001:** Average scores for the RE-AIM dimensions and the scores for each 33-item indicator as used in the current study.

Dimension	Definition	Indicator	Percentage (%)
Reach (R)	The proportion and representativeness of individuals willing to participate in a given intervention.	1. Described Target Population	100
		2. Demographic & behavioral information	100
		3. Recruitment Strategies	94.1
		4. Inclusion & exclusion criteria	94.1
		5. Method to identify the target population	76.5
		6. Sample size	88.2
		7. Participation rate	82.4
			Average: 90.8
Efficacy/Effectiveness (E)	The influence of an intervention on important outcomes, including potential negative effects, quality of life, and economic outcomes	1. Design/Conditions	100
		2. Efficacy, Effectiveness, Translational?	100
		3. Measure of the primary outcome	100
		4. Results (shortest assessment)	94.1
		5. Intent-to-treat or present at FU	35.3
		6. Imputation procedure	76.5
		7. Measure of robustness across subgroups	47.1
		8. Measure of short-term attrition	23.5
			Average: 72.1
Adoption (A)	The proportion and representativeness of locations and intervention staff willing to initiate and adopt an intervention	1. Description of intervention location	88.2
		2. Description of delivery staff	64.7
		3. Method to identify target delivery agent	47.1
		4. Level of expertise of delivery agent	58.8
		5. Delivery staff participation rate	11.8
		6. Organizational spread	5.9
		7. Measures of cost of adoption	5.9
			Average: 40.3
Implementation (I)	How consistently various elements of an intervention are delivered as intended by staff, and the time and cost of the intervention	1. Intervention frequency	88.2
		2. Extent protocol delivered as intended (%)	11.8
		3. Participant attendance/completion rates	82.4
		4. Measures of cost	0
			Average: 45.6
Maintenance (M)	The extent to which participants make and maintain a behavior change and the sustainability of a program or policy in the setting in which it was intervened	1. Follow-up assessment (3- or 6-months)	70.6
		2. Attrition	35.3
		3. Is the program still in place?	0
		4. Was the program institutionalized?	0
			Average: 26.5

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
