# Peer review of "Internal and External Validity of Social Media and Mobile Technology-Driven HPV Vaccination Interventions: Systematic Review Using the Reach, Effectiveness, Adoption, Implementation, Maintenance (RE-AIM) Framework"

_vaccines, 2021, doi:10.3390/vaccines9030197_

Round 1

Reviewer 1 Report

  1. I recommend the authors to provide citation for the two HPV reviews authors mentioning in the line no 97
  2. In line no 115: authors mentioned “Articles were excluded if they were cross-sectional studies “. Is all the articles selected in this study are longitudinal?
  3. The article is little confusing whether it is an review article showing the role of RE-AIM or it is an original research article using the RE-AIM for the analysis of articles related to HPV vaccination intervention.
  4. The authors needs to explain how their review adds knowledge to the HPV vaccination intervention using RE-AIM.
  5. Race /ethnicity plays a role in population-based studies. The authors need to discuss how RE-AIM framework will be useful in race-specific studies.
  6. The authors need to rewrite the manuscript in clear manner so that it is easy to understand for the readers.

Author Response

Reviewer #1 Comments

  1. I recommend the authors to provide citation for the two HPV reviews authors mentioning in the line no 97

Thank you for this comment. We have provided citation. Please see line 101

  1. In line no 115: authors mentioned “Articles were excluded if they were cross-sectional studies “. Is all the articles selected in this study are longitudinal?

Thank you very much for this comment. Intervention studies are the only articles we included in this review. We have clarified that in the inclusion criteria. Please see line 117

  1. The article is little confusing whether it is an review article showing the role of RE-AIM or it is an original research article using the RE-AIM for the analysis of articles related to HPV vaccination intervention.

Thanks for this comment. No, it is not an original article. From lines 100 to line 105 we stated that this is a systematic review. The lines read “Unlike the two previous reviews on HPV vaccine interventions [35, 36], our current systematic review was structured to determine the translation potential or external validity of published HPV vaccination intervention studies by determining the extent to which those studies reported information across all five of the RE-AIM framework dimensions. We provided recommendations for future research based on these findings.” 

  1. The authors needs to explain how their review adds knowledge to the HPV vaccination intervention using RE-AIM.

Thank you very much for this comment. This review contributes to HPV vaccination interventions by asking researchers to report all necessary components (RE-AIM framework) in HPV interventions so HPV studies can be replicated easily, and the findings translated into health practice. We believe these are how our reviewing is contributing to the knowledge.

(a) In the introduction section, we stated the gap in the literature concerning HPV vaccination intervention and the importance of using RE-AIM framework. Then we stated how the use of the RE-AIM framework will help address the gap. This information is presented in line 53 to line 93.

(b) Additionally, we explained our contributions to knowledge in the discussion, lines 389 to 399.

(c) We have made changes to the implications of the study. In the changes, we have explained that the review has significant implications for future publications and future HPV intervention studies. See lines 401 to 420

(d) Finally, our conclusion reiterates that both internal and external validities are important, and both should be considered in publications. See lines 422 to 432.

  1. Race /ethnicity plays a role in population-based studies. The authors need to discuss how RE-AIM framework will be useful in race-specific studies.

We appreciate the fact that the reviewer is concerned about race/ethnicity and how RE-AIM be useful. We are equally passionate about health disparities and race. However, as the review is not original primary research, we rely on the set of indicators provided in the reviewed manuscripts to evaluate existing publications. Unfortunately, addressing specific race and ethnic group is beyond the scope of this review. That said, a translational researcher who is using the RE-AIM framework to design HPV vaccination intervention or any original study is encouraged to address race/ethnic group. The Reach dimension in the RE-AIM framework is used to address research population including race and ethnicity. To draw attention to race/ethnicity, we have specified the proportion of reviewed studies that described their target population in terms of race/ethnicity as a specific demographic variable. Please see lines 208 and 210.

  1. The authors need to rewrite the manuscript in clear manner so that it is easy to understand for the readers.

We have substantially revised and rewritten the manuscript to improve clarity.

Reviewer 2 Report

The authors present a critical review of parameters used in the 17 selected publication to increase vaccine uptake and recommended dose completion.

The most common indicators reported are reach and efficacy/effectiveness whereas the external validity dimensions which include adoption, implementation, and maintenance were underreported. The conclusion is that without such information the translation of those intervention studies into practice will be difficult.

One major recommendation is to incorporated mobile technology which may increase vaccine uptake and recommended dose completion.

All this is convincing and warrants publication.

However, a reader not familiar with the technical terms used in the manuscript will have difficulty to follow. It is recommended to include a short paragraph which explains these terms or to explain them when mentioned first. Examples are uptake, maintenance, penetration, reach, external validity, etc.

The section 4.4 Implications and recommendations is a mix of recommendations for future studies as well as suggestions to improve acceptance of vaccination. Since the study is designed to improve publications, which investigate acceptability of vaccination both aspects should be discussed separately. 

Reviewer 3 Report

I agree that there is add on value of the current systematic review with the current RE-AIM approach, but I do have comments that are mainly related to the ‘gap’ between the analysis and the subsequent recommendations.

Specific comments

Title: do we need the ‘Evaluation of the’ in the title ? and perhaps using ‘the’ Reach, ….

I miss some data and reflections on the targeted population, are these the children, adolescents, young adults or parents or rather non-specific ? Are these tools sex-specific (male/female) ? the approach suggest that these aspects are considered in the analysis, but I was unable to retrieve this (reach dimension indicators).

College students are suggested, but no clear age category, and to what extent is it sufficiently early ?

I miss information on the ‘technical’ approaches taken: social media and mobile technology-driven techniques are quite diverse. Only limited information on this (study intervention assessed) aspect is mentioned in the implementation section (4.1.4.). You might consider to add study related information as a supplement (target ‘audience, type of intervention, assessment tools)

The implications and recommendations are valuable but

  1. I was not able to retrieve the information on the parents’ targeting
  2. The same holds true on the facebook claim: it is likely correct, but the results should be better reflected the subsequent recommendations
  3. The relevance of external validity: I assume that the potential to translate HPV vaccine research findings refers to the social media ? can you make this statement clearer ?

Round 2

Reviewer 3 Report

effective revision, suggest to accept